

# Transcriptome sequencing of *Verticillium dahliae* from a cotton farm reveals positive correlation between virulence and tolerance of sugar-induced hyperosmosis

Jin Li[1,*], Juan Pei[1,*], Yuanyuan Liu[1], Wenwen Xia[1], Fengfeng Cheng[1], Wenhui Tian[1], Zhongping Lin[2], Jianbo Zhu[1] and Aiying Wang[1]

[1] College of Life Science, Shihezi University, Shihezi, China
[2] College of Life Sciences, Peking University, Beijing, China
[*] These authors contributed equally to this work.

Corresponding author
Aiying Wang, way-sh@126.com

## ABSTRACT

*Verticillium dahliae* causes disease symptoms in its host plants; however, due to its rapid variability, *V. dahliae* is difficult to control. To analyze the reason for this pathogenic differentiation, 22 *V. dahliae* strains with different virulence were isolated from a cotton farm. The genetic diversity of cotton varieties make cotton cultivars have different *Verticillium wilt* resistance, so the Xinluzao 7 (susceptible to *V. dahliae*), Zhongmian 35 (tolerant), and Xinluzao 33 (resistant) were used to investigate the pathogenicity of the strains in a green house. Vegetative compatibility groups (VCGs) assays, Internal Transcribed Spacer (ITS) PCR, and pathogenicity analysis showed that SHZ-4, SHZ-5, and SHZ-9 had close kinship and significantly different pathogenicity. Transcriptome sequencing of the three strains identified 19 of 146 unigenes in SHZ-4_vs_ SHZ-5, SHZ-5_vs_ SHZ-9, and SHZ-4_vs_ SHZ-9. In these unigenes, three proteinase and four polysaccharide degrading hydrolases were found to be associated with the pathogenicity. However, due to a number of differentially expressed genes in the transport, these unigenes not only played a role in nutrition absorption but might also contribute to the resistance of sugar-induced hyperosmosis. Moreover, the tolerance ability was positively related to the pathogenicity of *V. dahliae*. This resistance to sugar-induced hyperosmosis might help *V. dahliae* to access the nutrition of the host. The pathogenicity of *V. dahliae* correlated with the resistance of sugar-induced-hyperosmosis, which provides clues for the cultivation of *V. dahliae* resistant varieties.

## INTRODUCTION

To stand and face various biotic and abiotic stresses, plants have evolved a wide spectrum of mechanisms to defend themselves against these stresses, especially pathogen infection. The mechanisms in response to pathogen attack are divided into two groups, pre-existing

and induced. The pre-existing mechanisms mainly involve physical and chemical barriers for protecting plants from pathogen infestation at the first line, including the plant cuticle, cell wall, and antimicrobial compounds (*Pieterse et al., 2009*; *Wang et al., 2019*). Salicylic acid (SA) crosstalking with auxin, ethylene, and jasmonates (JA) is crucial for the inducible response of plants to pathogens infection including systemic acquired resistance and induced systemic resistance (*Pieterse et al., 1998*; *Lopez, Bannenberg & Castresana, 2008*; *Vlot, Dempsey & Klessig, 2009*). In the plants response to *Verticillium dahliae* infection, many genes that are mostly involved in lignin biosynthesis and phenylpropanoid metabolic pathway are induced (*Li et al., 2019b*; *Yang et al., 2019*; *Zhang et al., 2019c*). The increasing of these gene expressions could confer resistance to the Verticillium wilt (VW) in cotton. For cotton varieties, the *Gossypium barbadense* possess innate resistance to VW and premium fiber quality, but low-fiber productivity upon infection (*Sun et al., 2013*). Thus, the development of new cotton cultivars possessing VW resistance by means of traditional breeding and transgenic strategies based on the pathogenic mechanism of *V. dahliae* are the most practical and cost effective method to manage cotton VW.

*Verticillium dahliae* is a fungal plant pathogen with wide global distribution and the ability to infect a number of important cash crops such as cotton, tomato, sunflower, and sugar beet, causing significant yield reduction (*Fradin & Thomma, 2006*; *Klosterman et al., 2009*). In response to an adverse environment, it produces a dormant structure (the microsclerotia) that can survive many years in cultivated soil and its germination is induced by exudates of plant roots (*Cai et al., 2009*; *Klosterman et al., 2011*). Upon induction, hyphae of *V. dahliae* cover the plant root surface and penetrate the root epidermal cells of susceptible plants (*Klimes & Dobinson, 2006*). Thereafter, the hyphae proliferate in the vascular system, which causes severe wilting of plants (*Huisman, 1982*; *Sanei et al., 2005*). Given the economic losses of cash crop production as a result of *V. dahliae* infection, understanding the mechanism of its pathogenicity is an urgent requirement.

*V. dahliae* causes apparent disease symptoms, which often include wilt and chlorosis, plant stunting, and vascular discoloration. Moreover, pathogenicity varied between hosts (*Fradin & Thomma, 2006*). Due to its broad host range, *V. dahliae* was assumed to have large genetic diversity. Thereafter, an increasing number of vegetative compatibility assays and molecular research illustrated that *V. dahliae* indeed had significant genetic diversity (*Bao et al., 1998*; *Korolev et al., 2001*; *Rowe, 1995*; *Baroudy et al., 2019*). In the study by *Göre (2007)*, 101 filtered isolates were assigned to four vegetative compatibility groups (VCGs), the pathogenicities of which ranged from weak to strong. *Korolev et al. (2001)* identified 62 isolates from Spain and 49 from Israel, which were classified into four VCG groups via VCG assays. Among each group, the virulence of different isolates ranged from weak to strong. Previously, strains from cotton had been classified into two groups based on symptoms (defoliation and non-defoliation) (*Schnathorst & Mathre, 1966*). *Daayf, Nicole & Geiger (1995)* further classied the *V. dahliae* into four groups via their vegetative compatibility and pathogenicity on cotton. Recently, *Dung et al. (2019)*, using genotyping by sequencing, PCR assays for mating-type and pathogenic race, vegetative compatibility group (VCG) tests, and aggressiveness assays, identified the genetic diversity of *V. dahliae*. The diversity at the population level was attributed to the ability for rapid genetic variation, which

called for the development of varieties that are resistant to *V. dahliae* infection due to the emergence of newly resistant strains.

Today, as a result of the development of omics technology, many of the genes that are responsible for the pathogenicity were identified. *Chu et al. (2015)* identified 106 secreted proteins from *V. dahliae* under nutrition stress. These proteins are involved in cell wall degradation, the scavenging of and stress response to reactive oxygen species (ROS), lipid effectors, protein metabolism, carbohydrate metabolism, electron–proton transport, and energy metabolism. *Chen et al. (2016)* investigated the exoproteome via cotton-containing medium induction, where the identified 271 secreted proteins were enriched in carbohydrate hydrolyses and carbohydrate-active (CAZymes) involved in pectin and cellulose degradation pathways. These proteins play central roles in the pathogenicity. Hydrolases involved in plant cell wall degradation were regarded as criteria proteins for the production of disease symptoms and pathogenicity (*King et al., 2011*; *Glass et al., 2013*; *Kubicek, Starr & Glass, 2014*; *Zhang et al., 2019a*). Knockdown of *sucrose non-fermenting 1* (*VdSNF1*) reduced expression of a number of PCWDEs and strongly impaired pathogenicity (*Tzima et al., 2011*). In addition, many genes were identified that play a role in pathogenicity, such as *VMK1*, *VdPKAC1*, *VdSge1*, and *VdMFS* that mainly function in adhesion and penetration (*Luo et al., 2014*). However, these findings cannot explain the virulence differentiation of the *V. dahliae* sufficiently. The DNA rearrangement and lineage-specific (LS) genomic regions are related to virulence and niche adaptation, and the major generators are transposable elements (TEs) (*De Jonge et al., 2013*). This promotes the rapid development of genes that are relevant to the host adaptation and virulence (*Klosterman et al., 2011*; *De Jonge et al., 2013*; *Amyotte et al., 2012*). *Chen et al. (2017)* confirmed that horizontal gene transfer from *Fusarium* to Vd991 significantly improved the ability for host adaptation.

Although studies on the pathogenicity and variation of *V. dahliae* have achieved considerable progress, the variation mechanism and key genes responsible for the different virulence of strains remain unclear. The following key problems remain: 1. The genetic background of *V. dahliae* species that were compared was not consistent—the majority of studies investigated *V. dahliae* collected from different regions; 2. *V. dahliae* possesses the ability for rapid variation. Therefore, the genes that play the main role in the pathogenicity differentiation remain unknown. The completion of *V. dahliae* genome sequencing enables the accurate study of the variation mechanism (*Chen et al., 2017*). To investigate the key variation gene between *V. dahliae* strains with high close genetic relationship and different pathogenicities, strains were collected from a cotton field of the Shihezi region, with the aim to reduce differences in the genetic background. VCGs and specific marker PCR were used to assure kinship. Virulence testing of all strains on three cotton varieties was used to analyze the pathogenicity differences of isolates. Through genetic relationship and pathogenicity differentiation, three isolates were selected for transcriptome sequencing with the aim to seek the key gene responsible for pathogenicity differentiation.

## MATERIALS & METHODS

### Molecular identification of *V. dahliae*

Field experiments was approved by Xinjiang Han, a personal farmer. To collect the *V. dahliae* strains with close relationship, infected cotton plants were collected from a field that the Verticillium wilt was serious in recent decades in the Shihezi region, Xinjiang province (E 86.04, N 44.30). The stems were separated into 0.5–1 cm slices, after which they were disinfected via $HgCl_2$ for 10–15 min. PDA medium was used to culture *V. dahliae*. The CTAB method was used to extract the DNA of *V. dahliae*. The ITS sequence was obtained via ITS1/ITS4 universal primer PCR. PCR was conducted using the following protocol: f 95 °C for 10 min; 30 cycles of 95 °C for 45 s, 56 °C for 30 s, 72 °C for 30 s, and 72 °C for 10 min. All ITS sequences were cloned into pMD19-T vector and sequenced. The Neighbor-joining model was used to construct the phylogenetic tree by ClustalW of the software Mega 5.0, using the default cutoff.

### Vegetative Compatibility Groups (VCGs) assay

All *V. dahliae* strains were used to obtain the Nit mutant strains. The process was conducted according to the method described in *Joaqium & Rowe (1990)*. All obtained Nit mutant strains were used to assess the vegetative compatibility relationships of *V. dahliae* strains. The experimental procedure followed the method described in *Joaqium & Rowe (1991)*. All the experiments were repeated three times.

### Pathogenicity assay on cotton plants

Xinluzao 7 (susceptible to the pathogen), Zhongmian 35 (tolerant), and Xinluzao 33 (resistant) were used to investigate the pathogenicity of *V. dahliae* in a green house of key laboratory of Shihezi university. The conidial concentration was estimated via hemocytometer and adjusted to $1 * 10^7$ CFU/ml. The experimental procedure and calculation of disease index were conducted according to *Zhu et al. (2013)*. All experiments were repeated three times. Variance analysis was conducted using SPSS v17.0 (SPSS Inc., Chicago, IL, USA).

### Transcriptome sequencing of *V. dahliae*

The SHZ-4, SHZ-5, and SHZ-9 strains was cultured in the Czapek liquid medium for 14 days at the conditions of 25 °C, dark, and 180 rpm/min. The spore suspension was abtained through filtering the hyphae off. The spore suspension was then centrifuged to collect the concentrated spores. The total RNA were extracted using the Fungal RNA miniprep kit (Biomiga, San Diego, CA, USA) according to the manufacturer's instructions. Total RNA was sent to the Sangon Biotech Cooperation (Shanghai, China) for RNA examination and sequencing. The sequencing platform was a HiSeq-2500-Pe-125 of Illumina. The total raw data was processed by the FASTQC software with default parameters. Cutadapt software used $O = 10$, min_len = 35, a = adaptor sequence. Prinseq software used the following parameter definitions: trim_qual_left = 20, trim_qual_right = 20, trim_qual_window = 10, trim_qual_step = 1, and the Blast+ with an $e$-value = 1e−10, num_threads = 40, min_len = 35, to obtain clean reads. The transcriptome was assembled by the Trinity

software (http://trinityrnaseq.github.io/) with min_contig_length set to 200. The longest transcript was considered as a unigene. All unigenes were successfully annotated with a similarity >30% and an *e*-value = 1e−5. Blastn (NCBI blast2.2.28+) was used against the NT database. For the unigenes BlastX (NCBI blast2.2.28+) was used against NR, SwissProt, and TrEMBL databases. For CDD, COG/KOG, and PFAM annotations, the rpsblast software (NCBI blast2.2.28+) was used. GO annotation was obtained according to the annotated result of both SwissProt and TrEMBL. The KEGG Automatic Annotation Server of KAAS was used for KEGG annotation. The RSEM/bowtie program (version 1.2.8; Madison, WI, USA) with a cutoff defined at $v = 2$ software was used for the mapping and the RSeQC (http://rseqc.sourceforge.net/) was used for the statistic of the mapping with default settings. The expression level of unigenes (Fragment Per Kilo bases per Million mapped Reads, FPKM) was calculated by the RSEM/bowtie program with default settings. Since the transcriptome sequencing had no biological repetition, the analysis of differential unigene expression was conducted in reference to the Audic method (*Audic & Claverie, 1997*). The DESeq and edgeR programs were used to identify the differential expression of unigenes and the cutoff was defined at $p \leq 0.01$, logFC $\geq 2$, and FDR $< 0.001$. The figure was merged by the Photoshop_CS3_SC_V1.3 (Adobe System Incorporation; San Jose, CA, USA).

## Validation of RNA-Seq data by qRT-PCR

Six unigenes being selected randomly from the data that were identified in all SHZ-4_vs_ SHZ-5, SHZ-5_vs_ SHZ-9, and SHZ-4_vs_SHZ-9 DEG data were used to confirm the expression patterns of the Illumina RNA-Seq results by quantitative real-time PCR (qRT-PCR). The ROCHE LightCycler® 480 system (Salt Lake City, UT, USA) was used to determine the expression level of the selected genes and the SYBR Green Real-Time PCR Master Mix (KAPA Biosystems, Wilmington, MA, USA) was utilized in 10 μL reactions. The reaction system consisted of 2 ng template, 0.8 μL primers, and 5 μL master mixes. The PCR reactions were performed in a thermocycler using the following conditions: 5 min at 95 °C, 45 cycles of 10 s at 95 °C, 15 s at 60 °C, and 25 s at 72 °C. Primers were designed with Primer Premier software (Primer Premier v5.0; Premier Biosoft International, Palo Alto, CA, USA). The *β*-tubulin gene was used as reference. The experiments were repeated three times. To assess the correlation between transcriptome sequencing and quantitative Real-Time PCR (qRT-PCR), Pearson's coefficient was calculated using OriginPro 8.6 (OriginLab; Northampton, MA, USA). The figure was merged by the Photoshop_CS3_SC_V1.3 (Adobe System Incorporation; San Jose, CA, USA).

## RESULTS

### Molecular identification and Vegetative Compatibility Groups (VCGs) of *V. dahliae*

The ITS sequences of the isolates were obtained via universal primer ITS PCR. Molecular phylogenetic tree analysis indicated differences in these strains (the ITS sequences of all strains could be found in the Datasets S1). All 22 isolates could be classified into 11 groups (Fig. 1). The SHZ-4, SHZ-5, SHZ-6, SHZ-9, and SHZ-11 strains were distributed in the
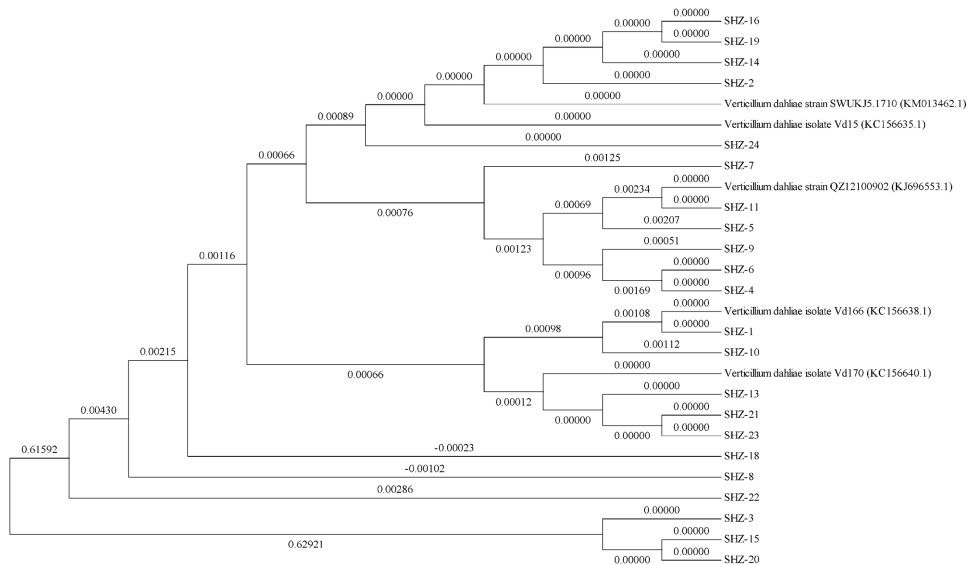

**Figure 1** **The phylogenetic tree analysis based on the ITS sequences of *V. dahliae*.**

same branch, and have a close relationship with KJ696553.1. In these isolates, both SHZ-4 and SHZ-9 had a closer relationship with SHZ-5 in comparison to the others. Few of these isolates formed groups, such as SHZ-3, SHZ-15, and SHZ-20.

For further analysis the genctic relationship, all the strains were used to obtain the Nit mutant strains to do the VCG analysis. In total, 53 Nit mutants were obtained from 10 of 22 strains. The total numbers of Nit1, Nit M, and Nit 3 mutants were 39, 10, and 4, respectively (Table S1). The mutant strains were SHZ-2, SHZ-4, SHZ-5, SHZ-6, SHZ-8, SHZ-9, SHZ-11, SHZ-13, SHZ-18, and SHZ-21, respectively (Table 1). Three vegetative compatibility groups were found via mutant pairing. SHZ-4, SHZ-5, SHZ-6, SHZ-8, SHZ-9, and SHZ-11 isolates belong to one group (VCG1), which was consistent with the molecular phylogenetic tree analysis. This indicates that the isolates SHZ-4, SHZ-5, SHZ-6, SHZ-8, SHZ-9, and SHZ-11 had a closer relationship relative to the others.

## Pathogenicity differentiation test on the cotton hosts of *V. dahliae* strains

The pathogenicity type could be divided into the following three categories: strong (average disease index value ≥ 40), moderate (25 ≤ average disease index value <40), and weak (average disease index value <25). The strong pathogenicity category consisted of seven strains, which were SHZ-7, SHZ-9, SHZ-13, SHZ-16, SHZ-21, and SHZ-3 (Table 2). The average values of disease indexes of these isolates ranged from 51.20 to 43.52. Half of the strains had moderate pathogenicity and the disease index values of which ranged from 35.48 to 26.24. The strains that belong to the moderate pathogenicity were the dominant population. SHZ-19, SHZ-10, SHZ-4, SHZ-24, and SHZ-14 were classified as weakly pathogenic. The disease index values of the weak pathogenicity group ranged from 21.57 to 18.36. According to both ITS analysis and VCGs identification, the SHZ-4, SHZ-5, SHZ-6,

**Table 1 Vegetative compatibility test of Nit mutants.**

| Nit1 | NitM | | | | | | VCGs |
|------|------|------|------|------|------|------|------|
|      | SHZ-2 | SHZ-4 | SHZ-11 | SHZ-8 | SHZ-5 | SHZ-9 | |
| SHZ-2 | + | − | + | − | − | − | VCG2 |
| SHZ-13 | − | − | − | − | − | − | VCG3 |
| SHZ-4 | − | + | − | − | + | +/− | VCG1 |
| SNZ-18 | + | − | + | − | − | − | VCG2 |
| SHZ-5 | − | − | − | + | + | − | VCG1 |
| SHZ-6 | − | + | − | + | +/− | − | VCG1 |
| SHZ-21 | − | − | − | − | − | − | VCG3 |
| SHZ-8 | − | + | − | + | + | + | VCG1 |
| SHZ-9 | − | +/− | − | − | + | + | VCG1 |
| SHZ-11 | − | − | + | − | − | + | VCG1 |

**Notes.**
The " +" represent the interstrain pairings with testers successed to yield prototrophic growth at the line of mycelial contact.

SHZ-8, SHZ-9, and SHZ-11 had a close relationship. However, these strains had different pathogenicity. This phenomenon illustrates that *V. dahliae* has a large variation at the population level.

## Transcriptome sequencing results for different pathogenicities of strains

To investigate the reasons of the observed pathogenicity differentiation, the three strains SHZ-4 (weak), SHZ-5 (moderate), and SHZ-9 (strong) that had a close relationship and different pathogenicity, were used for transcriptome sequencing. Total RNA was extracted from the strains, and sequenced by the Illumina HiSeq 2500 platform. A total of 15.58 Gb of raw data were acquired. All clean reads of the three strains were used to assemble the transcriptome. A total of 19,700 unigenes, ranging from 201 to 16031 base pairs (bp) with 2,884 bp of N50, were obtained (Fig. 2A). 67.4% of the unigenes were annotated in at least one database (NR, NT, PFAM, CDD, KEGG, GO, Swissprot, TrEMBL, or KOG; Fig. 2B). In SHZ-4_vs_SHZ-5, a total of 1908 differentially expressed genes (DEGs) were identified, 693 and 1215 of which were up- and down-regulated, respectively (Table S2). 1673 DEGs (808 upregulated and 865 downregulated) were found in SHZ-5_vs_SHZ-9. In SHZ-4_vs_SHZ-9, 610 unigenes were upregulated and 400 were downregulated (Fig. 2C). According to Venn-iagram analysis, 146 unigenes were identified in all sample DEG data sets.

## Quantitative real-time-PCR validation of differentially expressed transcripts from RNA-Seq

Six unigenes were used to validate the expression patterns of the Illumina RNA-Seq results via qRT-PCR (Fig. 3A and Table S3). These genes were extracellular trypsin protease (*VTP1*), SET domain-containing protein (*VDAG_03353*), dicarboxylic amino acid permease (*VDAG_10467*), rhamnogalacturonate lyase (*VDAG_07119*), ABC transporter (*VDAG_05668*), and cytochrome b2 (*VDAG_07114*). According to the qRT-PCR results, the expression levels of all six unigenes were lower in SHZ-4 than in SHZ-5 and SHZ-9.
**Table 2 Pathogenicity differentiation test on cotton plants infected with *V. dahliae*.**

| *V. dahliae* | Disease index | | | Average value |
|---|---|---|---|---|
| | Xinluzao 7 | Zhongmian 35 | Xinluzao 33 | |
| SHZ-9 | 78.06 | 48.89 | 26.67 | 51.20 |
| SHZ-16 | 77.22 | 46.94 | 28.33 | 50.83 |
| SHZ-13 | 69.44 | 50.00 | 30.33 | 49.93 |
| SHZ-7 | 75.00 | 42.22 | 30.33 | 49.19 |
| SHZ-21 | 69.44 | 45.56 | 25.33 | 46.78 |
| SHZ-3 | 72.22 | 35.00 | 23.33 | 43.52 |
| SHZ-20 | 48.06 | 33.70 | 24.67 | 35.48 |
| SHZ-18 | 50.56 | 35.28 | 19.67 | 35.17 |
| SHZ-6 | 50.83 | 34.44 | 19.33 | 34.87 |
| SHZ-22 | 50.83 | 32.59 | 21.00 | 34.81 |
| SHZ-11 | 44.72 | 38.06 | 21.33 | 34.70 |
| SHZ-5 | 53.61 | 31.11 | 15.33 | 33.35 |
| SHZ-8 | 41.67 | 34.44 | 21.33 | 32.48 |
| SHZ-15 | 48.89 | 28.89 | 19.33 | 32.37 |
| SHZ-23 | 46.67 | 29.63 | 18.33 | 31.54 |
| SHZ-1 | 56.11 | 23.33 | 14.33 | 31.26 |
| SHZ-2 | 43.61 | 21.11 | 14.00 | 26.24 |
| SHZ-19 | 23.70 | 30.00 | 11.00 | 21.57 |
| SHZ-10 | 25.56 | 28.33 | 10.33 | 21.41 |
| SHZ-24 | 20.74 | 28.33 | 11.33 | 20.14 |
| SHZ-4 | 22.22 | 27.22 | 9.67 | 19.70 |
| SHZ-14 | 21.85 | 23.89 | 9.33 | 18.36 |

The transcription of extracellular trypsin protease, dicarboxylic amino acid permease, and ABC transporter was ordered from low to high in the following: SHZ-4, SHZ-5, and SHZ-9, which is consistent with their pathogenicity. Furthermore, the correlation between RNA-Seq and qRT-PCR was evaluated using the foldchange value. As shown in Fig. 3B, the qRT-PCR measurements were moderately correlated with the RNA-Seq results ($r = 0.77$, $R2 = 0.56$), which indicates that the RNA-Seq data were accurate and could be used for gene expression profile analysis of the cold temperature defense response.

## Gene ontology enrichment analysis of differentially expressed unigenes

All DEGs were analyzed via GO enrichment. The terms that were enriched in the SHZ-4_vs_SHZ-5, SHZ-5_vs_SHZ-9, and SHZ-4_vs_SHZ-9 mainly focused on transport, membrane component, and relevant enzymes. In SHZ-4_vs_SHZ-5, SHZ-5_vs_SHZ-9, and SHZ-4_vs_SHZ-9 DEG data sets, the transport term was most prevalent (Figs. S1–S3). These were involved in the organic acid transmembrane transporter activity, organic anion transmembrane transporter activity, substrate-specific transmembrane transporter activity, and organophosphate ester transport. Peptidase, pathogenesis, and hydrolase activity terms were enriched, all of which were related to pathogenicity. In SHZ-4_vs_ SHZ-5 and

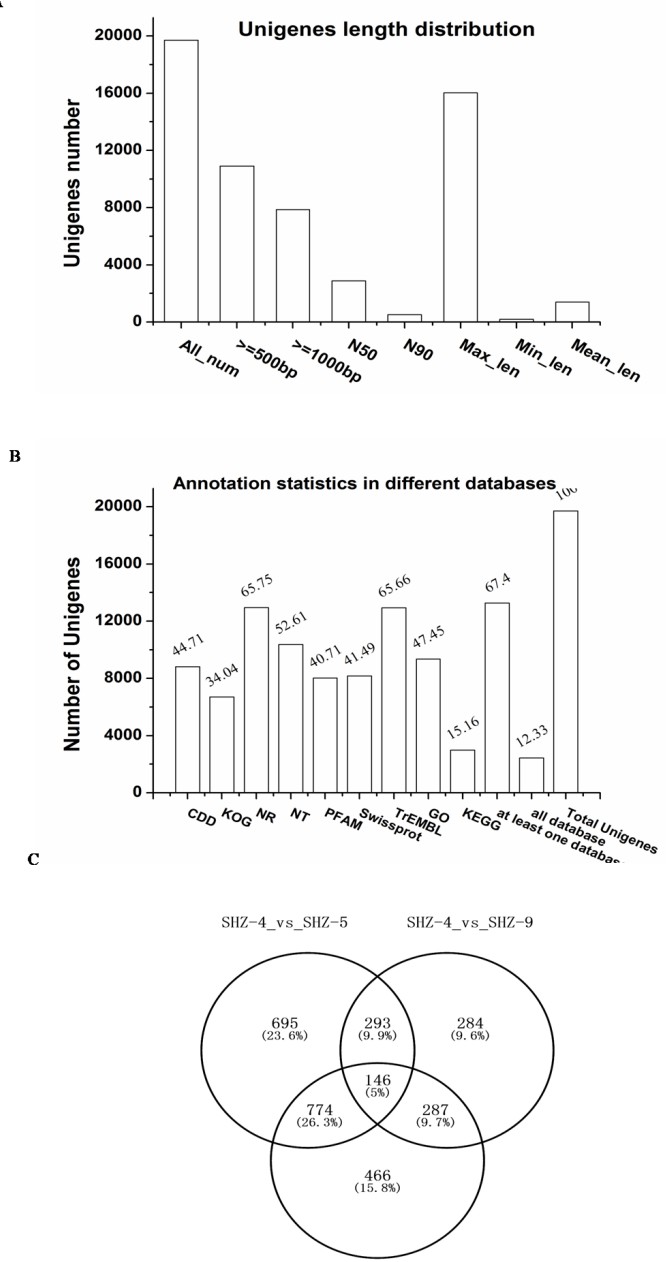

**Figure 2  Transcriptome assembly and annotation statistic.** (A) The unigenes length distribution satistic after transcriptome assembly; (B) the statistics of unigenes annotation in different databases; (C) Venn diagrams of comparisons between the DEGs of the strains.

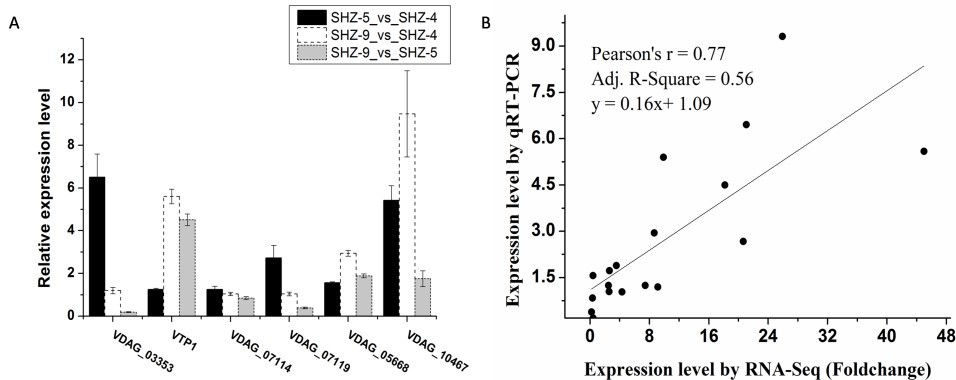

**Figure 3** **The unigenes expression level analyzed by the qRT-PCR.** (A) The relative expression level of the unigenes in different strains; (B) correlation analysis between RNA-Seq and qRT-PCR.

SHZ-5_vs_ SHZ-9, these terms were mainly concentrated in the sugar metabolic process. However, the protein degradation, carbohydrate relative process, and pathogenesis terms were enriched in SHZ-4_vs_SHZ-9.

## Transport proteins were vital in the pathogenicity

All unigenes on the transport were screened from the DEGs data set. There were 169 (9% of the total DEGs), 109 (11%), and 168 (10%) unigenes that were found in SHZ-4_vs_ SHZ-5, SHZ-4_vs_ SHZ-9, and SHZ-5_vs_ SHZ-9, respectively (Fig. 4A). In SHZ-4_vs_ SHZ-5, the transport proteins were mainly related to protein transporting (28, 16.57% of the total transport proteins), ion or inorganic salt transporting (30, 17.75%), carbohydrate transporting (47, 27.81%), other organic transporting (36, 21.30%), and amino acid transporting (18, 10.65%). Among these unigenes, the numbers of the transport proteins were 26, 19, 40, 30, and 6, respectively. The transport protein types, based on the transport substance type, were similar in SHZ-4_vs_SHZ-5, SHZ-4_vs_ SHZ-9, and SHZ-5_vs_SHZ-9. However, the transport proteins on the protein transporting with the largest percentage in SHZ-4_vs_ SHZ-9 differed from the SHZ-4_vs_SHZ-5 and SHZ-5_vs_ SHZ-9, and both carbohydrate transport unigenes and others organic transport unigenes provided the largest percentages in SHZ-4_vs_SHZ-5 and SHZ-5_vs_ SHZ-9, respectively. It is well known that the transport of secreted proteins is related to vesicle transport. Because the secreted proteins are related to pathogenicity, the unigenes involved in the vesicle transport were analyzed. There were eight, nine, and five unigenes on the vesicle transport in the SHZ-4_vs_SHZ-5, SHZ-4_vs_ SHZ-9, and SHZ-5_vs_ SHZ-9 in total. According to the expression level of these unigenes, only one unigene was screened from five unigenes in SHZ-5_vs_ SHZ-9.

There were twelve unigenes on the transport that extracted from the 146 unigenes data that were identified in all SHZ-4_vs_ SHZ-5, SHZ-5_vs_ SHZ-9, and SHZ-4_vs_SHZ-9 DEG data. In these unigenes, four unigenes (Lactose permease, Alpha-glucosides permease, Sugar transporter and L-fucose transporter) had the function of transporting carbohydrates (Fig. 4B). The expression level of all four unigens was highest in the SHZ-5. The expression

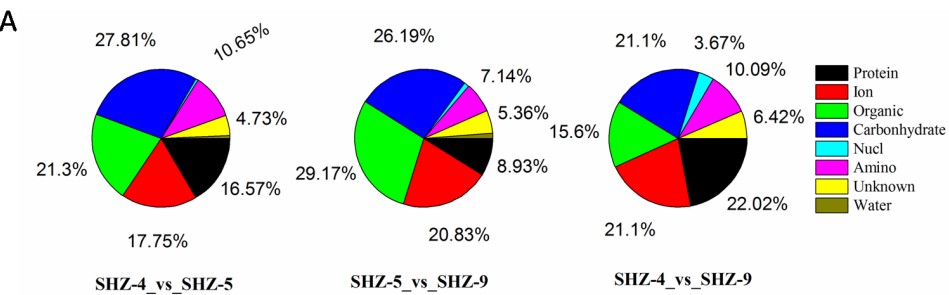

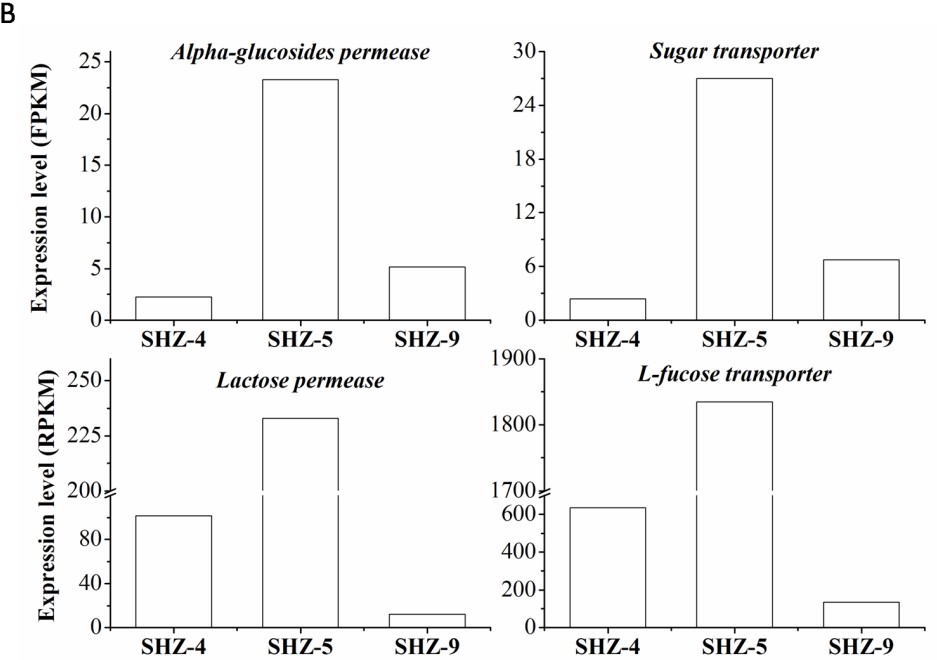

**Figure 4** **Unigenes on different substance transport.** (A) Unigenes proportion on different substance transport. (B) Expression level of genes on the carbonhydrates.

level of Lactose permease, Alpha-glucosides permease and Sugar transporter were lowest in the SHZ-4.

## Key genes responsible for pathogenicity differentiation

Unigenes (146) that were identified in all SHZ-4_vs_SHZ-5, SHZ-5_vs_SHZ-9, and SHZ-4_vs_SHZ-9 DEG data were used to analyze the expression mode (Fig. 5 and Table S4). Based on the pathogenicity characteristic of the isolates, 19 unigenes were extracted from the 146 unigenes. The expression level of these 19 unigenes was lower in the SHZ-4 than in the SHZ-5 and SHZ-9. Among these 19 unigenes, the expression levels of seven unigenes showed the same order as the FPKM value from low to high in the SHZ-4, SHZ-5, and SHZ-9 isolates. These seven unigenes were metalloproteinase, integral membrane protein, inorganic pyrophosphatase, alkaline proteinase, dicarboxylic amino acid permease, phosphate-repressible phosphate permease, and extracellular trypsin

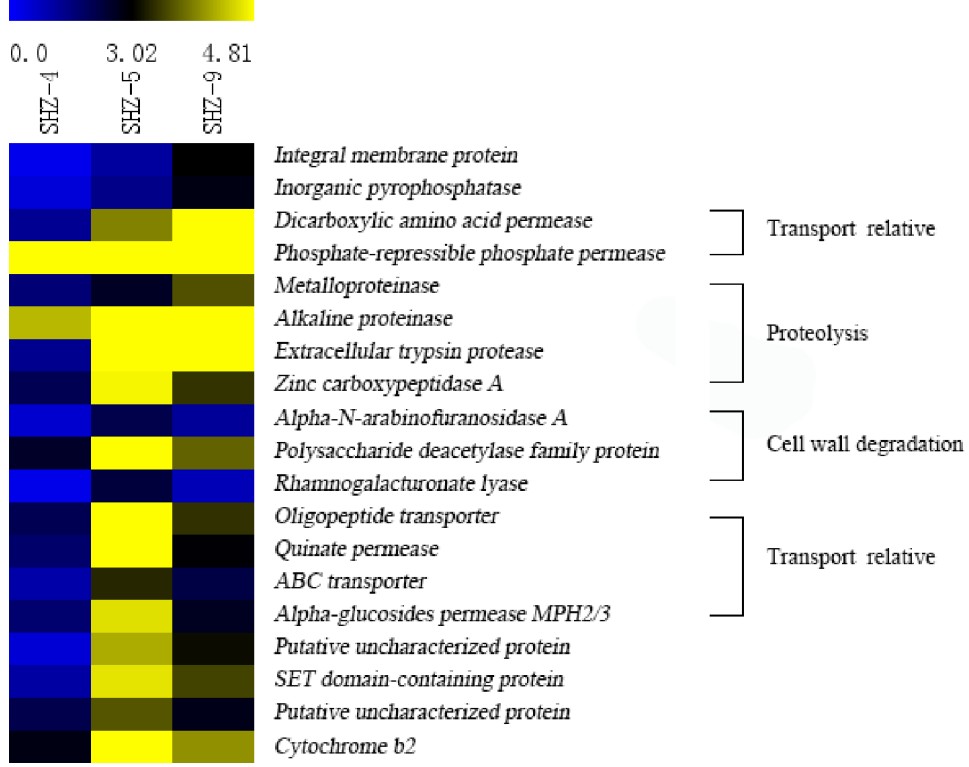

**Figure 5 Expression levels of 19 unigenes selected according to the pathogenicities of SHZ-4, SHZ-5, and SHZ-9.** The FPKM value was used as expression level value.

protease. Metalloproteinase, alkaline proteinase, and extracellular trypsin protease belonged to the secretary proteins and can be used as virulence factors responsible for pathogenicity. The expression levels of the remaining 12 unigenes differed from those of the seven unigenes, and the expression levels of the unigenes in SHZ-5 was higher than in SHZ-9. Among the 12 unigenes, four had a function in host cell component degradation, i.e., Alpha-N-arabinofuranosidase A, polysaccharide deacetylase family protein, zinc carboxypeptidase A, and rhamnogalacturonate lyase. Additional, four unigenes on the transport were filtered. These were oligopeptide transporter, ABC transporter, quinate permease, and alpha-glucosides permease MPH2/3.

## Sugar-induced hyperosmotic tolerance positive correlation with the virulence

The sugar-induced-hyperosmotic resistances of three *V. dahliae* isolates were analyzed by changing the sugar consistence (Fig. 6). The result illustrates that the stronger pathogenicity of *V. dahliae* yielded a higher resistance to hyperosmosis. The sugar tolerance limits of SHZ-4 and SHZ-9 were 50 g/L and 150 g/L, respectively, due to the growth inhibition of the hypha under these conditions. However, the SHZ-9 could maintain normal growth at a sugar concentration of 200 g/L. This suggests that a link should exist between pathogenicity and resistance to sugar hyperosmotic stress. It is widely accepted that hyperosmosis of

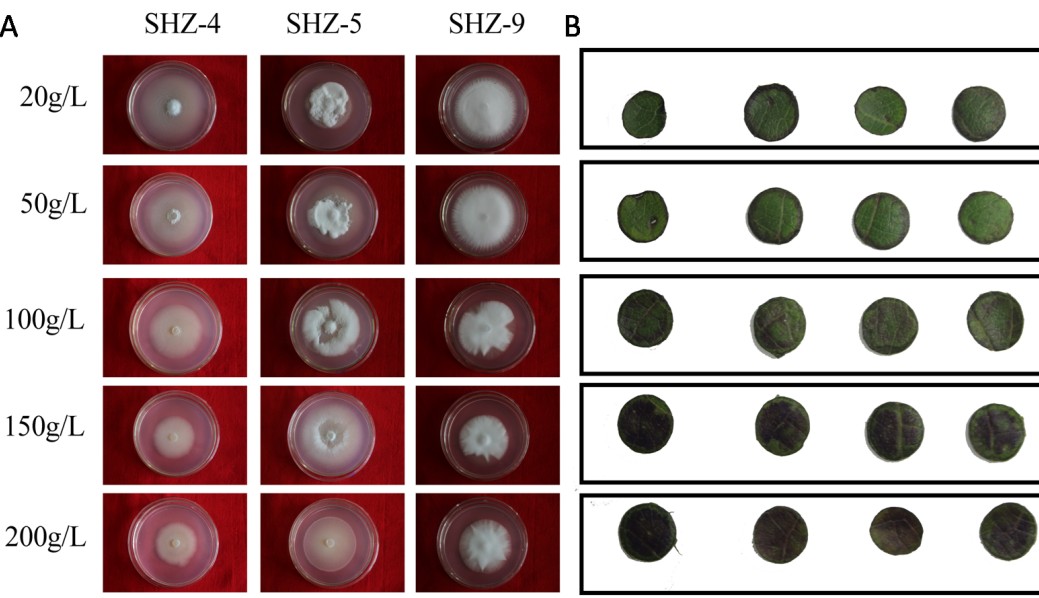

**Figure 6 Identification of *V. dahliae* strains and cotton leaves that are resistant to sugar-induced-hyperosmosis.** (A) The sugar-induced-hyperosmotic resistances of three *V. dahliae* isolates; (B) the cotton leaf resistance to sugar-induced-hyperosmosis.

intercellular liquid could not only leach moisture from the vessel into the hyperosmotic area via free diffusion, but could also help *V. dahliae* to scavenge nutrients from the host cell. To further investigate this, Zhongmian 35 was used to analyze the resistance to sugar-induced-hyperosmosis, which simulated the processes caused by *V. dahliae* infection. Round leaf tissue was obtained by placing samples in PDA medium with different sugar concentrations, similar to *V. dahliae* treatment. The result indicates that brown spots were produced in the round leaf tissue 5 days later. Moreover, higher sugar concentration led to a higher number of brown spots. Under conditions if high sugar concentration, these brown spots were produced in the leaf tissue, unlike in low sugar concentration stress, where the brown spots were produced at the edge of the leaf tissue. This was similar to the symptoms caused by *V. dahliae*, where brown spots would emerge in the leaves of infected plants. Integration of the *V. dahliae* sugar-induced-hyperosmosis resistance with cotton symptoms during hyperosmosis conditions shows that the symptoms caused by *V. dahliae* could, at least, be partly attributed to the hyperosmosis produced by the degradation process.

# DISCUSSION

## Cell-wall degrading enzymes underlying pathogenicity differentiation

In this study, 22 isolates of *V. dahliae* were obtained from a cotton field. Through ITS and VCG analysis, three isolates were found to have relatively close kinship, especially for SHZ-4 and SHZ-9. However, the isolates showed considerable differences in pathogenicity,

with virulence ranging from weak to strong with the order of SHZ-4, SHZ-5, and SHZ-9. A number of genes were found via transcriptome sequencing. The genes related to protein and carbohydrate degradation and transporting should be related to pathogenicity. The effectors as well as pectin and cellulose degrading enzymes were important for pathogenicity (*Buchner, Nachmias & Burstein, 1982*; *Buchner, Burstein & Nachmias, 1989*; *Davis, Low & Heinstein, 1998*; *Mansoori, Milton & Smith, 2010*; *Meyer, Slater & Dubery, 1994*; *Nachmias, Buchner & Burstein, 1985*). Genome comparison indicated that the *V. dahliae* genome contains more genes for cell-wall degrading enzymes (CWDEs) than other fungi (*Klosterman et al., 2011*). Seven genes belonged to the CWDEs in this study, which are likely responsible for the pathogenic differentiation of *V. dahliae*. However, the genes of endopolygalacturonase (*PG*), exopolygalacturonase (*PGX*), and glycoside hydrolase exerted no significant difference on the expression level. Accordingly, although these genes are important for the pathogenicity (*Liu et al., 2017a*), they were not the key effectors underlying pathogenic differentiation. A study reported evidence that the double mutant of *PG1* and *PGX6* only reduced the virulence in tomato (*Bravo Ruiz, Di Pietro & Roncero, 2016*).

## Transport proteins implicated in pathogenicity differentiation

Transport proteins played central role in the biological metabolism and regulation. Examples were carbohydrate transport for the energy metabolism, ion transport for metal-enzymes and osmotic balance, and synthetic protein transport for types of metabolisms or regulation. A large number of unigenes on the transport (about 10% of the total DEGs) were screened out. These genes were implicated in many biological processes, for example protein transport, ion transport, carbohydrate transport, and other organic transport. Therefore, many transport genes related to the pathogenic differentiation of *V. dahliae* were not found in previous studies.

Almost all transport proteins were mediated by vesicular trafficking, which included secreted proteins and intracellular proteins. In this study based on the GO annotation, most of the transport proteins belonged to intracellular protein transport and only two genes were involved in the transport of secreted proteins. Many genes on the transport of secreted protein that are responsible for pathogenesis have been identified, e.g., *MoSso1*, *MoSec22*, and *MoVamp7* in *Magnaporthe oryzae* (*Song et al., 2010*; *Dou et al., 2011*; *Giraldo et al., 2013*), *FgSso1*, *FgVam7*, and *FgVps39* in *Fusarium graminearum* (*Li et al., 2017*), and *VdSec22* and *VdSso1* in *V. dahliae* (*Wang et al., 2018a*). Many carbohydrate hydrolysis enzymes were secreted by VdSec22 and VdSso1 and were mediated by vesicular transport (*Wang et al., 2018a*). These results indicate that the genes of CWDEs mainly resulted in different protein levels, which could explain why few DEGs related to CWDEs were found. For the intracellular proteins transport, these unigenes were implicated in pathogenic differentiation, and they seemed to be not directly involved in pathogenicity. However, *Vac1p*, a vesicular transport protein for the intracellular protein transport from *Candida albicans*, was required for both virulence and development (*Franke et al., 2006*). Vac1p was important for metal ion resistance, such as $Cu^{2+}$, $Zn^{2+}$, and $Ni^{2+}$ (*Franke et al., 2006*). The result illustrates that the unigenes for the intracellular protein transport were not directly

related to pathogenicity. The gene *Vps34* encodes a key protein for vacuolar protein transport, which is involved in the secretion of aspartyl proteinases, metal ion resistance, and fungistatic compounds (*Kitanovic et al., 2005*). A intracellular protein transported by the vesicular transport system should be more closely related to the pathogenicity than the vesicular transport genes on the intracellular protein transport (e.g., ion channels).

The carbohydrates, salts, and metal ions, and other organic molecules were most important for *V. dahliae* nutrition. Most of the nutrition was absorbed via degradation of host-cells by the CWDEs secreted by *V. dahliae*. With regard to the nutrient components, iron ions have been studied in many pathogens. Iron is an essential nutrient for nearly all living organisms, including eukaryotes and prokaryotes (*Kaplan & Kaplan, 2009*). Siderophores are most important for obtaining iron from the extremely low available iron of host cells, which could contribute to the iron homeostasis in the pathogen (*Haas, 2003*). However, the ABC-type $Fe^{3+}$ transporter was only found in our study. The expression level of ABC-type Fe3+ transporter was highest in the SHZ-9 and lowest in the SHZ-4, which was positive correlation with the virulence of the three strains. Iron homeostasis could promote resistance to oxidative stress, through which the iron ion can affect the virulence of the pathogen (*Gómez & Nosanchuk, 2003*).

In addition to the nutrient consumption, transport proteins could play a role in other process. With regard to virulence genes, the CWDEs could degrade the macromolecules of host cells into small molecules. However, these small molecules greatly improved the osmotic pressure of the intercellular liquid. Therefore, improving the effectiveness of plant polygalacturonase-inhibiting proteins (PGIPs) in reducing fungal polygalacturonase (PG) activity could enhance cotton resistance to *V. dahliae* (*Liu et al., 2018*; *Liu et al., 2017b*). The winter embolism recovery of the walnut tree, where air bubbles in xylem vessels are cleared by an osmotic force produced via high-level *PIP2* expression and the increase of sucrose from the conversion of starch-to-sugar corroborated this (*Sakr et al., 2003*). Likewise, the *V.dahliae* could affect the xylem vessels of host plants to produce cavitations and starch hydrolysis, and vessel occlusion was considered relevant to wilt symptoms (*Trapero et al., 2018*). Hyperosmosis of the intercellular liquid would yield a detrimental effect on the surrounding of the host cell, which could lead to loss of moisture and small molecules. The crop cultivars that could resist the *V. dahliae* were associated with lignin biosynthesis, such as tomato (*Hu et al., 2019*) and cotton (*Li et al., 2019a*). The strengthening of cell walls might protect the host cell against the hyperosmotic harm. The effect of hyperosmosis should be integrated into the symptoms caused by *V. dahliae*. This physiological phenomenon indicated that *V. dahliae* had the ability to resist hyperosmosis and a positive correlation will likely exist between virulence and the ability for hyperosmotic tolerance. The *VdSsk2* gene, a mitogen-activated protein kinase kinase kinases (MAPKKKs) homologous, was important in calcium signaling in fungi. Deletion of *VdSsk2* reduced the resistance to osmotic stress and virulence of *V. dahliae* (*Yu et al., 2019*). Ectopic expression of *GhSNAP33* enhancing the tolerance of yeast cells to oxidative and osmotic stresses and *GhSNAP33*-deficient cotton being susceptible to *V. dahliae* infection (*Wang et al., 2018b*), illustrated the relation of hyperosmotic resistance to Verticillium wilt. In *Zhang et al. (2019b)* research, the *RD21-7* (responsive to desiccation 21-7) expression was induced by

heat, cold and *V. dahliae* infection and over-expression of *GhRD21-7* enhanced resistance to *V. dahliae* in cotton. The *GhRD21-7* gene conferring resistance to *V. dahliae* in cotton might be associates with the tolerance of hyperosmotic stress, because of the cold stress inhibiting the water uptake by root and causing the intracellular water losses. Therefore, preventing the intracellular water losses of host plants would confer resistance to *V. dahliae* in theory. The hypothesis could be verified by *Wang et al.*'s (*2019*) research that the increased suberin of root cell wall improved the tolerance to *V. dahliae* in *Arabidopsis*. It could be inferred that high resistance to hyperosmotic stresses could confer resistance to *V. dahliae*. In *Dong et al. (2019)*, gene silencing of a ATP-binding cassette (ABC) transporter F family member 5 (*ABCF5*) gene in *Gossypium hirsutum* reduced the resistance to *V. dahliae*. The *ABCF5* gene might improve the tolerance to hyperosmotic stresses to enhance the immunity of cotton to *V. dahliae* infection. Therefore, the resistance of *V. dahliae* to hyperosmosis could affect its pathogenicity. In summary, transport-related DEGs could be implicated in the pathogenicity differentiation of *V. dahliae* by changing the resistance to hyperosmosis.

## CONCLUSIONS

Different *V. dahliae* strains possess different pathogenicity as a result of their large genetic variation. This variation was mainly encoded within CWDEs and transport-related genes. It is widely accepted that the CWDEs determined the source of nutrients for *V. dahliae* and the organic substances of the host cells degraded by CWDEs would induce hyperosmosis of the intercellular liquid. Hyperosmosis not only induced dehydration and death of the host cell, but also contributed to the nutrient absorption by *V. dahliae* due to its high hyperosmosis tolerance of strains with that strong pathogenicity. In summary, both CWDEs and transport-related genes are associated with the pathogenicity of *V. dahliae*. However, whether these transport-related genes are associated with the hyperosmosis tolerance requires further research.

### Funding

This work was funded by the transgenic major project of cotton (2016ZX08011-004) and the National Natural Science Foundation of China (31360053). The funders had no role in study design, data collection and analysis, decision to publish, or preparation of the manuscript.

### Grant Disclosures

The following grant information was disclosed by the authors:
Transgenic major project of cotton: 2016ZX08011-004.
National Natural Science Foundation of China: 31360053.

### Competing Interests

The authors declare there are no competing interests.

## Author Contributions

- Jin Li performed the experiments, analyzed the data, contributed reagents/materials/analysis tools, prepared figures and/or tables, authored or reviewed drafts of the paper, approved the final draft.
- Juan Pei, Yuanyuan Liu and Wenwen Xia performed the experiments, analyzed the data, contributed reagents/materials/analysis tools, prepared figures and/or tables, approved the final draft.
- Fengfeng Cheng and Wenhui Tian performed the experiments, prepared figures and/or tables, approved the final draft.
- Zhongping Lin and Jianbo Zhu conceived and designed the experiments, authored or reviewed drafts of the paper, approved the final draft.
- Aiying Wang conceived and designed the experiments, analyzed the data, authored or reviewed drafts of the paper, approved the final draft.

## Field Study Permissions

The following information was supplied relating to field study approvals (i.e., approving body and any reference numbers):

Field experiments were approved by Xinjiang Han, a personal farmer.

## Data Availability

Data is available at NCBI GEO, accession numbers: GSE131467.

## Supplemental Information

Supplemental information for this article can be found online at http://dx.doi.org/10.7717/peerj.8035#supplemental-information.

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
