# Peer review of "Transcriptome sequencing of Verticillium dahliae from a cotton farm reveals positive correlation between virulence and tolerance of sugar-induced hyperosmosis"

_PeerJ, doi:10.7717/peerj.8035_

## Round 0.1 · original submission · Minor Revisions

Please address all of the reviewer's comments provided in your revision.

Reviewer 1 ·

Basic reporting

Generally the introduction is well written, except that I would wish to draw to the attention of the authors on the most current publications on cotton response mechanism to V. dahliae, this will help to enrich the introduction and makes it much better being the latest referenced journal is 2017, between then to date a lot has been done.
The Role of ABC promoters in enhancing resistance to V. dahliae infection (Dong et al., 2019)
Dong Q, Magwanga R, Cai X, Lu P, Nyangasi Kirungu J, Zhou Z, Wang X, Wang X, Xu Y, Hou Y, Wang K, Peng R, Ma Z, Liu F. 2019. RNA-Sequencing, Physiological and RNAi Analyses Provide Insights into the Response Mechanism of the ABC-Mediated Resistance to Verticillium dahliae Infection in Cotton. Genes 10:110. DOI: 10.3390/genes10020110.

Experimental design

Minor modifications are required in the material and methods section;
i. Briefly describe the sample location, and why the site choice? Under the subsection “Molecular identification of V. dahliae”
ii. Validation of RNA-Seq Data by qRT-PCR: the authors has stated that “was used to determine the expression level of the selected genes and the SYBR Green Real-Time PCR Master Mix” provide a brief explanation on which criterion was adopted in the selection of the genes used in RT-qPCR validation”

Validity of the findings

The results are well described and supported with relevant literature citations, though few mistakes need to be corrected by the authors before submitting their final version
i. Line 226 Change “ gene otology” to “gene ontology”
ii. Line 239: a numerical term cannot begin the sentence, “All unigenes on the transport were screened from the DEGs data set. 169 (9% of the total…”
iii. Gene full name description should not be italicized unless the abbreviation form is given. Line 307 to 310 and line 314 to 317 in the description of the unigenes

Additional comments

The work is very interesting and will appeal to the wider audience in cotton science. The research problem is well laid out and the methods used are appropriate.

Annotated reviews are not available for download in order to protect the identity of reviewers who chose to remain anonymous.

·

Basic reporting

The standard of English for most of the report is good and forms a clear narrative.

In the abstract delete the word "apparent" in line 18. On line 25 I suggest "associated with" rather than "responsible for".

Experimental design

The methods for DNA extraction, sequencing, generation of nit mutants, and pathogenicity assay all seem good and conform with best practice.
Some steps such as cloning ITS products for before sequencing and validation of RNA-seq data by qRT-PCR are excellent steps that are often missing from lesser manuscripts.

I have no comment on the transcriptomics

Validity of the findings

I analysed the attached data file of ITS sequences. There are some problems with the DNA sequences and their analysis:

SHZ-17 is not Verticillium but rather Neocosmospora rubicola (See locus2 tree)

SHZ-12 is not Verticillium but rather an Alternaria species (See locus3 tree)

This makes me very concerned for the analysis of Figure 1. I don’t see how such a tree could be calculated from this sequence data. I have attached my own tree with a very quick analysis as tree.pdf it is rather different.

Looking at the raw sequence data it appears that nearly all the variation is at the ends of the sequence. This strongly suggests that the ends are messy low-quality sequence and should be removed or validated against raw chromatogram data. (see sequence picture). Were ITS primers or the plasmid primers used to sequence it insert?
The sequence data and its analysis needs to be greatly improved.

I have no comment on the transcriptomics

---

## Round 0.2 · Minor Revisions

Please address the request of Reviewer 2 that two of the DNA sequences should be resequenced.

Reviewer 1 ·

Basic reporting

good

Experimental design

good

Validity of the findings

good

Additional comments

accept

·

Basic reporting

.

Experimental design

.

Validity of the findings

.

Additional comments

I thank the authors for their revisions and response to the review.
However in the current dataset the DNA sequences of
SHZ-12 matches an Alternaria species and
SHZ-17 matches a Neocosmospora species
This can be verified by analysing the sequences at https://blast.ncbi.nlm.nih.gov

I acknowledge the authors response showing the gel photos indicating bands consistent with Verticillium dahliae, so perhaps there was a mix up with these sequences and other isolates.

In this case SHZ-17 and SHZ-12 should be re-sequenced.

---

## Round 0.3 · accepted · Accept

Thank you for addressing the reviewer's comments.